# Zero-Shot Semantic Segmentation

**Maxime Bucher**
valeo.ai
maxime.bucher@valeo.com

**Tuan-Hung Vu**
valeo.ai
tuan-hung.vu@valeo.com

**Matthieu Cord**
Sorbonne Université
valeo.ai
matthieu.cord@lip6.fr

**Patrick Pérez**
valeo.ai
patrick.perez@valeo.com

## Abstract

Semantic segmentation models are limited in their ability to scale to large numbers of object classes. In this paper, we introduce the new task of zero-shot semantic segmentation: learning pixel-wise classifiers for *never-seen* object categories with zero training examples. To this end, we present a novel architecture, ZS3Net, combining a deep visual segmentation model with an approach to generate visual representations from semantic word embeddings. By this way, ZS3Net addresses pixel classification tasks where both seen and unseen categories are faced at test time (so called "generalized" zero-shot classification). Performance is further improved by a self-training step that relies on automatic pseudo-labeling of pixels from unseen classes. On the two standard segmentation datasets, Pascal-VOC and Pascal-Context, we propose zero-shot benchmarks and set competitive baselines. For complex scenes as ones in the Pascal-Context dataset, we extend our approach by using a graph-context encoding to fully leverage spatial context priors coming from class-wise segmentation maps.

## 1 Introduction

Semantic segmentation has achieved great progress using convolutional neural networks (CNNs). Early CNN-based approaches classify region proposals to generate segmentation predictions [17]. FCN [28] was the first framework adopting fully convolutional networks to address the task in an end-to-end manner. Most recent state-of-the-art models like UNet [37], SegNet [3], DeepLabs [10, 11], PSPNet [50] are FCN-based. An effective strategy for semantic segmentation is to augment CNN features with contextual information, e.g. using atrous/dilated convolution [10, 46], pyramid context pooling [50] or a context encoding module [47].

Segmentation approaches are mainly supervised, but there is an increasing interest in weakly-supervised segmentation models using annotations at the image-level [33, 34] or box-level [13]. We propose in this paper to investigate a complementary learning problem where part of the classes are missing altogether during the training. Our goal is to re-engineer existing recognition architectures to effortlessly accommodate these never-seen, a.k.a. unseen, categories of scenes and objects. No manual annotations or real samples, only unseen labels are needed during training. This line of works is usually coined *zero-shot* learning (ZSL).

ZSL for image classification has been actively studied in recent years. Early approaches address it as an embedding problem [1, 2, 6, 8, 16, 24, 32, 36, 39, 42, 43, 48]. They learn how to map image data and class descriptions into a common space where semantic similarity translates into spacial proximity. There are different variants in the literature on how the projections or the similarity measure are

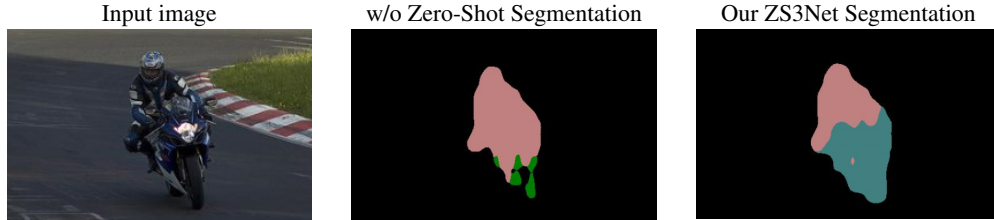

| Input image | w/o Zero-Shot Segmentation | Our ZS3Net Segmentation |

Figure 1: **Introducing and addressing zero shot semantic segmentation**. In this example, there are no 'motorbike' examples in the training set. As a consequence, a supervised model (middle) fails on this object, seeing it as a mix of the seen classes person, bicycle and background. With proposed ZS3Net method (right), pixels of the never-seen motorbike class are recognized.

computed: simple linear projection [1, 2, 6, 16, 24, 36], non-linear multi-modal embeddings [39, 43] or even hybrid methods [8, 32, 42, 48].

Recently, [7, 25, 45] proposed to generate synthetic instances of unseen classes by training a conditional generator from the seen classes. There are very few extensions of zero-shot learning to other tasks than classification. Very recently [4, 14, 35, 51] attacked object detection: unseen objects are detected while box annotation is only available for seen classes. As far as we know, there is no approach considering zero-shot setting for segmentation.

In this paper, we introduce the new task of zero-shot semantic segmentation (ZS3) and propose an architecture, called ZS3Net, to address it: Inspired by most recent zero shot classification apporaches, we combine a backbone deep net for image embedding with a generative model of class-dependent features. This allows the generation of visual samples from unseen classes, which are then used to train our final classifier with real visual samples from seen classes and synthetic ones from unseen classes. Figure 1 illustrates the potential of our approach for visual segmentation. We also propose a novel self-training step in a relaxed zero-shot setup where unlabelled pixels from unseen classes are already available at training time. The whole zero-shot pipeline including self-training is coined as ZS5Net (ZS3Net with Self-Supervision) in this work.

Lastly, we further extend our model by exploiting contextual cues from spatial region relationship. This strategy is motivated by the fact that similar objects not only share similar properties but also similar contexts. For example, 'cow' and 'horse' are often seen in fields while most 'motorbike' and 'bicycle' show in urban scenes.

We report evaluations of ZS3Net on two datasets (Pascal-VOC and Pascal-Context) and in zero-shot setups with varying numbers of unseen classes. Compared to a ZSL baseline, our method delivers excellent performances, which are further boosted using self-training and semantic contextual cues.

## 2 Zero-shot semantic segmentation

### 2.1 Introduction to our strategy

Zero-shot learning addresses recognition problems where not all the classes are represented in the training examples. This is made possible by using a high-level description of the categories that helps relate the new classes (the *unseen classes*) to classes for which training examples are available (the *seen classes*). Learning is usually done by leveraging an intermediate level of representation, which provides semantic information about the categories to classify.

A common idea is to transfer semantic similarities between linguistic identities from some suitable text embedding space to a visual representation space. Effectively, classes like 'zebra' and 'donkey' that share a lot of semantic attributes are likely to stay closer in the representation space than very different classes, 'bird' and 'television' for instance. Such a joint visual-text perspective enables statistical training of zero-shot recognition models.

We address in this work the problem of learning a semantic segmentation network capable of discriminating between a given set of classes where training images are only available for a subset of it. To this end, we start from an existing semantic segmentation model trained with a supervised loss on seen data (DeepLabv3+ in Fig. 2). This model is limited to trained categories and, hence, unable to recognize new unseen classes. In Figure 1-(middle) for instance, the person (a seen class) is

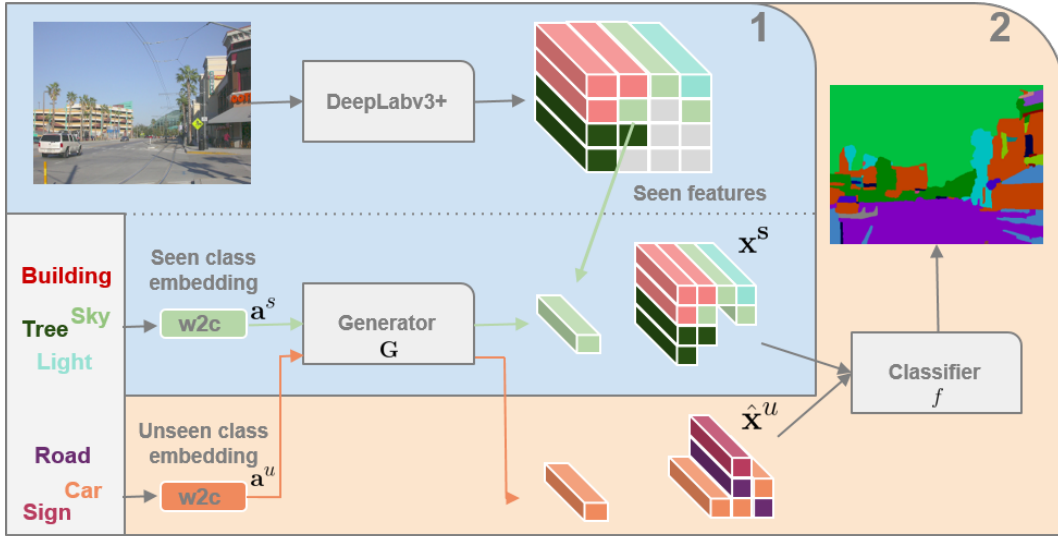

Figure 2: **Our deep ZS3Net for zero-shot semantic segmentation.** The figure is separated into two parts by colors corresponding to: (1) training generative model and (2) fine-tuning classification layer. In (1), the generator, conditioned on the word2vec (w2c) embedding of seen classes' labels, learns to generate synthetic features that match real DeepLab's ones on seen classes. Later in (2), the classifier is trained to classify real features from seen classes and synthetic ones from unseen classes. At run-time, the classifier operates on real DeepLab features stemming from both types of classes.

correctly segmented unlike the motorbike (an unseen class) whose pixels are wrongly predicted as a mix of 'person', 'bicycle' and 'background'.

To allow the semantic segmentation model to recognize both seen and unseen categories, we propose to generate synthetic training data for unseen classes. This is obtained with a generative model conditioned on the semantic representation of target classes. This generator outputs pixel-level multi-dimensional features that the segmentation model relies on (blue zone 1 in Fig. 2).

Once the generator is trained, many synthetic features can be produced for unseen classes, and combined with real samples from seen classes. This new set of training data is used to retrain the classifier of the segmentation network (orange zone 2 in Fig. 2) so that it can now handle both seen and unseen classes. At test time, an image is passed through the semantic segmentation model equipped with the retrained classification layer, allowing prediction for both seen and unseen classes. Figure 1-(right) shows a result after this procedure, with the model now able to delineate correctly the motorbike category.

## 2.2   Architecture and learning of ZS3Net

We denote the set of all classes as $\mathcal{C} = \mathcal{S} \cup \mathcal{U}$, with $\mathcal{S}$ the set of seen classes and $\mathcal{U}$ the set of unseen ones, where $\mathcal{S} \cap \mathcal{U} = \emptyset$. Each category $c \in \mathcal{C}$ can be mapped through word embedding to a vector representation $\mathbf{a}[c] \in \mathbb{R}^{d_a}$ of dimension $d_a$. In the experiments, we will use to this end the 'word2vec' model [30] learned on a dump of the Wikipedia corpus (approx. 3 billion words). This popular embedding is based on the skip-gram language model which is learned through predicting the context (nearby words) of words in the target dictionary. As a result of this training strategy, words that frequently share common contexts in the corpus are located in close proximity in the embedded vector space. In other words, this semantic representation is expected to capture geometrically the semantic relationship between the classes of interest.

In FCN-based segmentation frameworks, the input images are forwarded through an encoding stack consisting of fully convolutional layers, which results in smaller feature maps compared to the original resolution. Effectively, logit prediction maps have small resolutions and often require an additional up-sampling step to match the input size [10, 11, 28]. In the current context, with a slight abuse of notation, we attach each spatial location on the encoded feature map to a pixel of input image down-sampled to a comparable size. We can therefore assign class labels to encoded features

and construct training data in the feature space. From now on, 'pixel' will refer to pixel locations in this down-sampled image.

**Definition and collection of pixel-wise data (step 0).** We start from DeepLabv3+ semantic segmentation model [11], pre-trained with a supervised loss on annotated data from seen classes. Based on this architecture, we need to choose suitable features, out of several feature maps that can be used independently for classification. Conversely, the classifier to be later fine-tuned must be able to operate on individual pixel-wise features. As a result, we choose the last $1 \times 1$ convolutional classification layer of DeepLabv3+ and the features it ingests as our classifier $f$ and targeted features $\mathbf{x}$ respectively.

Let the training set $\mathcal{D}_s = \{(\mathbf{x}_i^s, \mathbf{y}_i^s, \mathbf{a}_i^s)\}$ be composed of triplets where $\mathbf{x}_i^s \in \mathbb{R}^{M \times N \times d_x}$ is a $d_x$-dimensional feature map, $\mathbf{y}_i^s \in \mathcal{S}^{M \times N}$ is the associated ground-truth segmentation map and $\mathbf{a}_i^s \in \mathbb{R}^{M \times N \times d_a}$ is the class embedding map that associates to each pixel the semantic embedding of its class.

We note that $M \times N$ is the resolution of the encoded feature maps, as well as of the down-sampled image and segmentation ground-truth.[1] For the $K = |\mathcal{U}|$ unseen classes, no training data is available, only the category embeddings $\mathbf{a}[c]$, $c \in \mathcal{U}$.

On seen data, the DeepLabv3+ model is trained with full-supervision using the standard cross-entropy loss. After this training phase, we remove the last classification layer and only use the remaining network for extracting seen features, as illustrated in blue part (1) of Figure 1. To avoid supervision leakage from unseen classes [44], we retrain the backbone network on ImageNet [38] on seen classes solely. Details are given later in Section 3.1.

**Generative model (step 1).** Key to our approach is the ability to generate image features conditioned on a class embedding vector, without access to any images of this class. Given a random sample $\mathbf{z}$ from a fixed multivariate Gaussian distribution, and the semantic description $\mathbf{a}$, new pixel features will be generated as $\widehat{\mathbf{x}} = \mathbf{G}(\mathbf{a}, \mathbf{z}; \mathbf{w}) \in \mathbb{R}^{d_x}$, where $\mathbf{G}$ is a trainable generator with parameters $\mathbf{w}$. Toward this goal, we can leverage any generative approach like GAN [18], GMMN [27] or VAE [22]. This feature generator is trained under supervision of features from seen classes.

We follow [7] and adopt the "Generative Moment Matching Network" (GMMN) [27] for the feature generator. GMNN is a parametric random generative process $\mathbf{G}$ using a differential criterion to compare the target data distribution and the generated one. The generative process will be considered as good if, for each semantic description $\mathbf{a}$, two random populations $\mathcal{X}(\mathbf{a}) \subset \mathbb{R}^{d_x}$ from $\mathcal{D}_s$ and $\widehat{\mathcal{X}}(\mathbf{a}; \mathbf{w})$ sampled with the generator have low *maximum mean discrepancy* (a classic divergence measure between two probability distributions):

$$L_{\text{GMMN}}(\mathbf{a}) = \sum_{\mathbf{x}, \mathbf{x}' \in \mathcal{X}(\mathbf{a})} k(\mathbf{x}, \mathbf{x}') + \sum_{\widehat{\mathbf{x}}, \widehat{\mathbf{x}}' \in \widehat{\mathcal{X}}(\mathbf{a}; \mathbf{w})} k(\widehat{\mathbf{x}}, \widehat{\mathbf{x}}') - 2 \sum_{\mathbf{x} \in \mathcal{X}(\mathbf{a})} \sum_{\widehat{\mathbf{x}} \in \widehat{\mathcal{X}}(\mathbf{a}, \mathbf{w})} k(\mathbf{x}, \widehat{\mathbf{x}}),$$

where $k$ is a kernel that we choose as Gaussian, $k(\mathbf{x}, \mathbf{x}') = \exp(-\frac{1}{2\sigma^2} \|\mathbf{x} - \mathbf{x}'\|^2)$ with bandwidth parameter $\sigma$. The parameters $\mathbf{w}$ of the generative network are optimized by Stochastic Gradient Descent [5].

**Classification model (step 2).** Similar to DeepLabv3+, the classification layer $f$ consists of a $1 \times 1$ convolutional layer. Once $\mathbf{G}$ is trained in step 1, arbitrarily many pixel-level features can be sampled for any classes, unseen ones in particular. We build this way a synthetic unseen training set $\widehat{\mathcal{D}}_u = \{(\widehat{\mathbf{x}}_j^u, y_j^u, \mathbf{a}_j^u)\}$ of triplets in $\mathbb{R}^{d_x} \times \mathcal{U} \times \mathbb{R}^{d_a}$. Combined with the real features from seen classes in $\mathcal{D}_s$, this set of synthetic features for unseen categories allows the fine-tuning of the classification layer $f$. The new pixel-level classifier for categories in $\mathcal{C}$ becomes $\widehat{y} = f(\mathbf{x}; \widehat{\mathcal{D}}_u, \mathcal{D}_s)$. It can be used to conduct the semantic segmentation of images that exhibit objects from both types of classes.

**Zero-shot learning and self-training.** Self-training is a useful strategy in semi-supervised learning that leverages a model's own predictions on unlabelled data to heuristically obtain additional pseudo-annotated training data [52]. Assuming that unlabelled images with objects from unseen classes are now available (this is thus a relaxed setting compared to pure ZSL), such a self-supervision can be mobilized to improve our zero-shot model. The trained ZS3Net (one gotten after step 2) can indeed be used to "annotate" these additional images automatically, and for each one, the top $p\%$ of the most confident among these pseudo-labels provide new training features for unseen classes. The semantic segmentation network is then retrained accordingly. We coin this new model ZS5Net for ZS3Net with Self-Supervision.

Note that there exists a connection between ZS5Net and transductive zero-shot learning [26, 40, 49], but that our model is not transductive. Indeed, under purely transductive settings, no data even unlabelled, is available at train time for unseen classes. Differently, our ZS5Net learns from a mix of labelled and unlabelled training data, and is evaluated on a different test set (effectively, a form of semi-supervised learning).

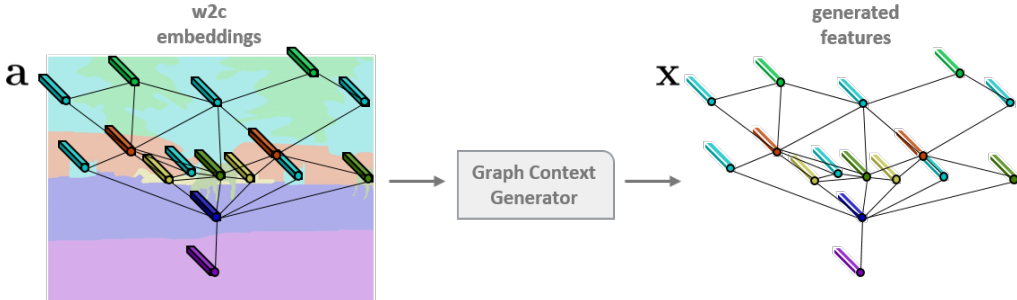

Figure 3: **Graph-context encoding in the generative pipeline**. The segmentation mask is encoded as an adjacency graph of semantic connected components (represented as nodes with different colors in the graph). Each semantic node is attached to its corresponding word2vec embedding vector. The generative process is conditioned on this graph. The generated output is also a graph with the same structure as the input's, except that attached to each output node is a generated visual feature. Best viewed in color.

**Graph-context encoding.** Understanding and utilizing contextual information is very important for semantic segmentation, especially in complex scenes. Indeed, by design, FCN-based architectures already encode context with convolutional layers. For more context encoding, DeepLabv3+ applies several parallel dilated convolutions at different rates. In order to reflect this mechanism we propose to leverage contextual cues for our feature generation. To this end, we introduce a graph-based method to encode the semantic context for complex scenes with lots of objects as ones in the Pascal-Context dataset.

In general, the structural object arrangement contains informative cues for recognition. For example, it is common to see 'dog' sitting on 'chairs' but very rare to have 'horses' doing the same thing. Such spatial priors are naturally captured by a form of relational graph, that can be constructed in different ways, *e.g.*, using manual sketches or semantic segmentation masks of synthetic scenes. What matters most is the relative spatial arrangement, not the precise shapes of the objects. As a proof of concept, we simply exploit true segmentation masks during the training of the generative model. It it important to note that the images associated to these masks are not used during training if they contain unseen classes.

A segmentation mask is represented by the adjacency graph $\mathcal{G} = (\mathcal{V}, \mathcal{E})$ of its semantic connected components: each node corresponds to one connected group of single class labels, and two such groups are neighbors if they share a boundary (hence being of two different classes). We re-design the generator to accept $\mathcal{G}$ as additional input using graph convolutional layers [23]. As shown in Figure 3, each input node $\nu \in \mathcal{V}$ is represented by concatenating its corresponding semantic embedding $\mathbf{a}_\nu$ with a random Gaussian sample $\mathbf{z}_\nu$. This modified generator outputs features attached to the nodes of the input graph.

## 3 Experiments

### 3.1 Experimental details

**Datasets.** Experimental evaluation is done on the two datasets: Pascal-VOC 2012 [15] and Pascal-Context [31]. Pascal-VOC contains $1,464$ training images with segmentation annotations of 20 object classes. Similar to [11], we adopt additional supervision from semantic boundary annotations [19] during training. Pascal-Context provides dense semantic segmentation annotations for Pascal-VOC 2010, which comprises $4,998$ training and $5,105$ validation images of 59 object/stuff classes.

**Zero-shot setups.** We consider different zero-shot setups varying in number of unseen classes, we randomly construct the 2-, 4-, 6-, 8- and 10-class unseen sets. We extend the unseen set in an incremental manner, meaning that for instance the 4- unseen set contains the 2-. Details of the splits are as follows:

Table 1: **Zero-shot semantic segmentation on Pascal-VOC.**

| $K$ | Model | Seen | | | Unseen | | | Overall | | | |
|---|---|---|---|---|---|---|---|---|---|---|---|
| | | PA | MA | mIoU | PA | MA | mIoU | PA | MA | mIoU | hIoU |
| | Supervised | – | – | – | – | – | – | 94.7 | 87.2 | 76.9 | – |
| 2 | Baseline | 92.1 | 79.8 | 68.1 | 11.3 | 10.5 | 3.2 | 89.7 | 73.4 | 44.1 | 6.1 |
| | ZS3Net | **93.6** | **84.9** | **72.0** | **52.8** | **53.7** | **35.4** | **92.7** | **81.9** | **68.5** | **47.5** |
| 4 | Baseline | 89.9 | 72.6 | 64.3 | 10.3 | 10.1 | 2.9 | 86.3 | 62.1 | 38.9 | 5.5 |
| | ZS3Net | **92.0** | **78.3** | **66.4** | **43.1** | **45.7** | **23.2** | **89.8** | **72.1** | **58.2** | **34.4** |
| 6 | Baseline | 79.5 | 45.1 | 39.8 | 8.3 | 8.4 | 2.7 | 71.1 | 38.4 | 33.4 | 5.1 |
| | ZS3Net | **85.5** | **52.1** | **47.3** | **67.3** | **60.7** | **24.2** | **84.2** | **54.6** | **40.7** | **32.0** |
| 8 | Baseline | 75.8 | **41.3** | **35.7** | 6.9 | 5.7 | 2.0 | 68.3 | 34.7 | 24.3 | 3.8 |
| | ZS3Net | **81.6** | 31.6 | 29.2 | **68.7** | **62.3** | **22.9** | **80.3** | **43.3** | **26.8** | **25.7** |
| 10 | Baseline | 68.7 | 33.9 | 31.7 | 6.7 | 5.8 | 1.9 | 60.1 | 26.9 | 16.9 | 3.6 |
| | ZS3Net | **82.7** | **37.4** | **33.9** | **55.2** | **45.7** | **18.1** | **79.6** | **41.4** | **26.3** | **23.6** |

| Models | Generalized eval. | Vanilla eval. |
|---|---|---|
| Baseline | 1.9 | 41.7 |
| ZS3Net | **18.1** | **46.2** |

Table 2: **Generalized- vs. vanilla ZSL evaluation**. Results are reported with mIoU metric on the 10-unseen split from Pascal-VOC dataset.

*Pascal-VOC:* 2-cow/motorbike, 4-airplane/sofa, 6-cat/tv, 8-train/bottle, 10-chair/potted-plant;
*Pascal-Context:* 2-cow/motorbike, 4-sofa/cat, 6-boat/fence, 8-bird/tvmonitor, 10-keyboard/aeroplane.

**Evaluation metrics.** In our experiments we adopt standard semantic segmentation metrics [28], i.e. pixel accuracy (PA), mean accuracy (MA) and mean intersection-over-union (mIoU). Similar to [44], we also report harmonic mean (hIoU) of seen and unseen mIoUs. The reason behind choosing harmonic rather than arithmetic mean is that seen classes often have much higher mIoUs, which will significantly dominate the overall result. As we expect ZS3 models to produce good performance for both seen and unseen classes, the harmonic mean is an interesting indicator.

**A zero-shot semantic segmentation baseline.** As a baseline, we adapt the ZSL classification approach in [16] to our task. To this end, we first modify the vanilla segmentation network, i.e. DeepLabv3+, to not produce class-wise probabilistic predictions for all the pixels, but instead to regress corresponding semantic embedding vectors. Effectively, the last classification layer of DeepLabv3+ is replaced by a projection layer which transforms $256-$channel features maps into $300-$channel word2vec embedding maps. The model is trained to maximize the cosine similarity between the output and target embeddings.

At run-time, the label predicted at a pixel is the one with the text embedding which the most similar (in cosine sense) to the regressed embedding for this pixel.

**Implementation details.** We adopt the DeepLabv3+ framework [11] built upon the ResNet-101 backbone [20]. Segmentation models are trained by SGD [5] optimizer using polynomial learning rate decay with the base learning rate of $7e^{-3}$, weight decay $5e^{-4}$ and momentum 0.9.

The GMMN is a multi-layer perceptron with one hidden layer, leaky-RELU non-linearity [29] and dropout [41]. In our experiments, we fix the number of hidden neurons to 256 and set the kernel bandwidths as $\{2, 5, 10, 20, 40, 60\}$. These hyper-parameters are chosen with the "zero-shot cross-validation procedure" in [7]. The input Gaussian noise has the same dimension as used w2c embeddings, namely 300. The generative model is trained using Adam optimizer [21] with the learning rate of $2e^{-4}$. To encode the graph context as described in Section 2.2, we replace the linear layers in GMMN by graph convolutional layers [23], with no change in other hyper-parameters.

## 3.2 Zero-shot semantic segmentation

We report in Tables 1 and 3 results on Pascal-VOC and Pascal-Context datasets, according to the three metrics. Instead of only evaluating on the unseen set (which does not show the strong prediction bias toward seen classes), we jointly evaluate on all classes and report results for seen, unseen, and all classes. Such an evaluation protocol is more challenging for zero-shot settings, known as "generalized

Table 3: **Zero-shot semantic segmentation on Pascal-Context.**

| K | Model | Seen | | | Unseen | | | Overall | | | |
|---|-------|------|------|------|--------|------|------|---------|------|------|------|
| | | PA | MA | mIoU | PA | MA | mIoU | PA | MA | mIoU | hIoU |
| | Supervised | – | – | – | – | – | – | 73.9 | 52.4 | 42.2 | – |
| 2 | Baseline | 70.2 | 47.7 | 35.8 | 9.5 | 10.2 | 2.7 | 66.2 | 43.9 | 33.1 | 5.0 |
| | ZS3Net | 71.6 | 52.4 | **41.6** | 49.3 | 46.2 | 21.6 | 71.2 | 52.2 | 41.0 | 28.4 |
| | ZS3Net + GC | **73.0** | **52.9** | 41.5 | **65.8** | **62.2** | **30.0** | **72.6** | **53.1** | **41.3** | **34.8** |
| 4 | Baseline | 66.2 | 37.9 | 33.4 | 9.0 | 8.4 | 2.5 | 62.8 | 34.6 | 30.7 | 4.7 |
| | ZS3Net | 68.4 | 46.1 | 37.2 | **58.4** | **53.3** | 24.9 | 67.8 | 46.6 | 36.4 | 29.8 |
| | ZS3Net + GC | **70.3** | **49.1** | **39.5** | 61.0 | 56.3 | **29.1** | **69.0** | **49.7** | **38.6** | **33.5** |
| 6 | Baseline | 60.8 | 36.7 | 31.9 | 8.8 | 8.0 | 2.1 | 55.9 | 33.5 | 28.8 | 3.9 |
| | ZS3Net | 63.3 | 38.0 | 32.1 | **63.6** | **55.8** | 20.7 | 63.3 | 39.8 | 30.9 | 25.2 |
| | ZS3Net + GC | **64.5** | **42.7** | **34.8** | 57.2 | 53.3 | **21.6** | **64.2** | **43.7** | **33.5** | **26.7** |
| 8 | Baseline | 54.1 | 24.7 | 22.0 | 7.3 | 6.8 | 1.7 | 49.1 | 20.9 | 19.2 | 3.2 |
| | ZS3Net | 51.4 | 23.9 | 20.9 | 68.2 | 59.9 | 16.0 | 53.1 | 28.7 | 20.3 | 18.1 |
| | ZS3Net + GC | **53.0** | **27.1** | **22.8** | **68.5** | **61.1** | **16.8** | **54.6** | **31.4** | **22.0** | **19.3** |
| 10 | Baseline | 50.0 | 20.8 | 17.5 | 5.7 | 5.0 | 1.3 | 45.1 | 16.8 | 14.3 | 2.4 |
| | ZS3Net | **53.5** | 23.8 | 20.8 | 58.6 | 43.2 | 12.7 | **52.8** | 27.0 | 19.4 | 15.8 |
| | ZS3Net + GC | 50.3 | **27.9** | **24.0** | **62.6** | **47.8** | **14.1** | 51.2 | **31.0** | **22.3** | **17.8** |

ZSL evaluation" [9]. In both Tables 1 and 3, we report in the first line the 'oracle' performance of the model trained with full-supervision on the complete dataset (including both seen and unseen).

**Pascal-VOC.** Table 1 reports segmentation performance on 5 different splits comparing the ZSL baseline and our approach ZS3Net. We observe that the embedding-based ZSL baseline (DeViSe [16]), while nicely performing on seen classes, produces much worse results for the unseen. We conducted additional experiments, adapting ALE [1] with $K = 2$. They yielded 68.1% and 4.6% mIoU for seen and unseen classes (harmonic mean of 8.6%), on a par with the DeViSe-based baseline.

Not strongly harmed by the bias toward seen classes, the proposed ZS3Net provides significant gains (in PA, MA and most importantly mIoU) on the unseen classes, e.g. +32.2% mIoU in the 2-split, similarly large gaps in other splits. As for seen classes, the ZS3Net performs comparably to the ZSL baseline, with slight improvement in some splits. Overall, we have favourable results on all the classes using our generative approach. The last column of Table 1 shows harmonic mean of the seen and unseen mIoUs, denoted as hIoU (%). As discussed in 3.1, the harmonic mean is a good indicator of effective zero-shot performance. Again according to this metric, ZS3Net outperforms the baseline by significant margins.

As mentioned in 2.2, our framework is agnostic to the choice of the generative model. We experimented a variant of ZS3Net based on GAN [45], which turned out to be on a par with the reported GMMN-based one. In our experiments, GMMN was chosen due to its better stability.

In all experiments, we only report results with the generalized ZSL evaluation. Table 2 shows the difference between this evaluation protocol and the common vanilla one. In the vanilla case, prediction scores of seen classes are completely ignored, only unseen scores are used to classify the unseen objects. As a result, this evaluation protocol does not show how well the models discriminate the seen and unseen pixels. In zero-shot settings, predictions are mostly biased toward seen classes given the strong supervision during training. To clearly reflect such a bias, the generalized ZSL evaluation is a better choice. We see that the ZSL baseline achieves reasonably good result using the vanilla ZSL evaluation while showing much worse figures with the generalized one. With ZS3Net, leveraging both real features from seen classes and synthetic ones from unseen classes to train the classifier helps reduce the performance bias toward the seen set.

The two first examples in Fig. 4 illustrate the merit of our approach. Model trained only on seen classes ('w/o ZSL') interprets unseen objects as background or as one of the seen classes. For example, 'cat' and 'plane' (unseen) are detected as 'dog' and 'boat' (seen); a large part of the 'plane' is considered as 'background'. The proposed ZS3Net correctly recognizes these unseen objects.

**Pascal-Context.** In Table 3, we provide results on the Pascal-Context dataset, a more challenging benchmark compared to Pascal-VOC. Indeed Pascal-Context scene pixels are densely annotated with

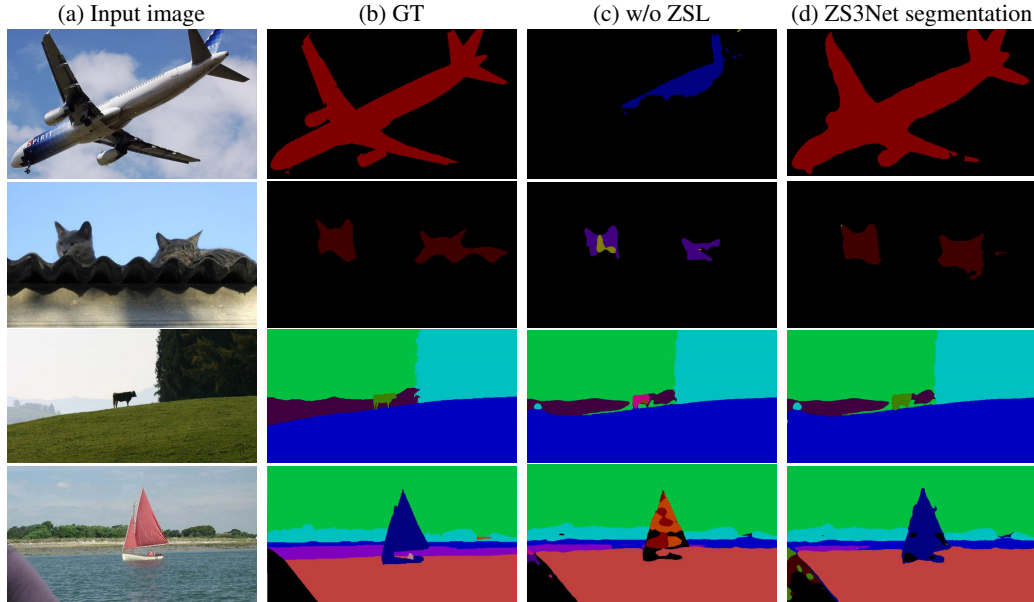

|  | (a) Input image | (b) GT | (c) w/o ZSL | (d) ZS3Net segmentation |

**Figure 4: Qualitative results on Pascal-VOC and Pascal-Context**. (a) Input image, (b) semantic segmentation ground-truth, (c) segmentation without zero-shot learning, (d) results with proposed ZS3Net. Unseen classes: plane cat cow boat; Some seen classes: dog bird horse dining-table. Best viewed in color.

Table 4: **Zero-shot with self-training**. ZS5Net results on Pascal-VOC and Pascal-Context datasets.

| K | Dataset | Seen | | | Unseen | | | Overall | | | |
|---|---------|------|------|------|--------|------|------|---------|------|------|------|
|   |         | PA   | MA   | mIoU | PA     | MA   | mIoU | PA      | MA   | mIoU | hIoU |
| 2 | VOC     | 94.3 | 85.2 | 75.7 | 89.5   | 89.9 | 75.8 | 94.2    | 85.9 | 75.8 | 75.7 |
|   | Context | 72.9 | 53.6 | 41.8 | 81.0   | 78.1 | 55.5 | 71.8    | 50.6 | 42.0 | 47.7 |
| 4 | VOC     | 93.9 | 84.8 | 74.0 | 57.5   | 62.9 | 53.0 | 92.4    | 80.9 | 69.8 | 61.8 |
|   | Context | 66.0 | 46.3 | 37.5 | 86.4   | 82.8 | 45.1 | 68.0    | 48.5 | 38.0 | 41.0 |
| 6 | VOC     | 93.8 | 82.0 | 71.2 | 68.3   | 61.2 | 53.1 | 92.1    | 75.8 | 66.1 | 60.8 |
|   | Context | 59.7 | 42.2 | 34.6 | 80.9   | 76.8 | 36.0 | 62.1    | 45.8 | 35.2 | 35.2 |
| 8 | VOC     | 92.6 | 77.8 | 68.3 | 68.2   | 62.0 | 50.0 | 90.2    | 71.9 | 61.3 | 57.7 |
|   | Context | 51.8 | 34.1 | 28.5 | 76.2   | 71.3 | 24.1 | 54.3    | 39.5 | 27.8 | 26.1 |
| 10| VOC     | 90.1 | 83.9 | 72.3 | 57.8   | 48.0 | 34.5 | 86.8    | 66.9 | 54.4 | 46.7 |
|   | Context | 46.8 | 32.3 | 27.0 | 70.2   | 57.1 | 20.7 | 49.5    | 36.4 | 26.0 | 23.4 |

59 object/stuff classes, compared to only a few annotated objects (most of the time 1-2 objects) per scene in Pascal-VOC. As a result, segmentation models universally report lower performance on Pascal-Context. Regardless of such difference, we observe similar behaviors from the ZSL baseline and our method. The ZS3Net outperforms the baseline by significant margins on all evaluation metrics. We emphasize the important improvements on the seen classes, as well as the overall harmonic mean of the seen and unseen mIoUs. We visualize in the two last rows of Figure 4 qualitative ZS3 results on Pascal-Context. Unseen objects, i.e. 'cow' and 'boat', while being wrongly classified as some seen classes without ZSL, can be fully recognized by our zero-shot framework.

As mentioned above, Pascal-Context scenes are more complex with much denser object annotations. We argue that, in this case, context priors on object arrangement convey beneficial cues to improve recognition performance. We have introduced a novel graph-context mechanism to encode such prior in Section 2.2. Proposed models enriched with this graph context, denoted as 'ZS3Net + GC' in Table 3, show consistent improvements over the ZS3Net models. We note that for semantic segmentation, the pixel accuracy metric (PA) is biased toward the more dominant classes and might suggest misleading conclusions [12], as opposed to IoU metrics.

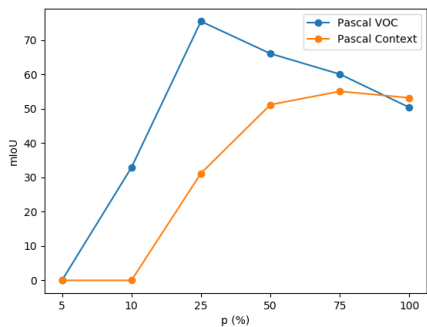

Figure 5: **Influence of parameter** $p$ **on ZS5Net**. Evolution of the mIoU performance as a function of percentage of high-scoring unseen pixels, on the 2-unseen classes split from Pascal-VOC and Pascal-Context datasets.

| (a) Input image | (b) GT | (c) ZS3Net | (d) ZS5Net |
|---|---|---|---|

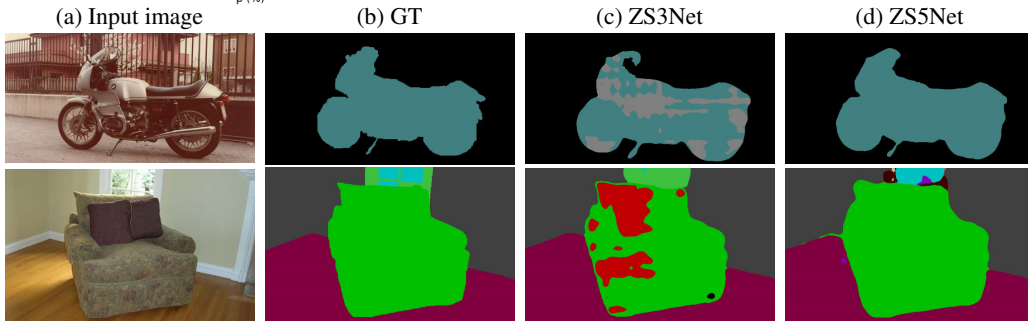

Figure 6: **Zero shot segmentation with self-training**. (a) Input image, (b) semantic segmentation ground-truth, (c) segmentation with ZS3Net, (d) result with additional self-training (ZS5Net). Unseen classes: motorbike sofa; Some seen classes car chair. Best viewed in color.

## 3.3 Zero-shot segmentation with self-training

We report performance of ZS5Net (ZS3Net with self-training) on Pascal-VOC and Pascal-Context in Table 4 (with different splits according to datasets). For performance comparison, the reader is referred to ZS3Net results in Tables 1 and 3. Through zero-shot cross-validation we fixed the percentage of high-scoring unseen pixels as $p = 25\%$ for Pascal-VOC and $p = 75\%$ for Pascal-Context. We show in Figure 5 the influence of this percentage on the final performance. In general, the additional self-training step strongly boosts the performance in seen, unseen and all classes. Remarkably, on the 2-unseen split in both datasets, the overall performance in all metrics is very close to the supervised performance (reported in the first lines of Tables 1 and 3). Figure 6 shows semantic segmentation results on Pascal-VOC and Pascal-Context datasets. On both cases, self-training helps to disambiguate pixels wrongly classified as seen classes.

## 4 Conclusion

In this work, we introduced a deep model to deal with the task of zero-shot semantic segmentation. Based on zero-shot classification, our ZS3Net model combines rich text and image embeddings, generative modeling and classic classifiers to learn how to segment objects from already seen classes as well as from new, never-seen ones at test time. First of its kind, proposed ZS3Net shows good behavior on the task of zero shot semantic segmentation, setting competitive baselines on various benchmarks. We also introduced a self-training extension of the approach for scenarios where unlabelled pixels from unseen classes are available at training time. Finally, a graph-context encoding has been used to improve the semantic class representation of ZS3Net when facing complex scenes.

## Footnotes

[1]Final softmax and decision layers operate in effect after logits are up-sampled back to full resolution.

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
