[Supplementary Material]

# Zero-Shot Semantic Segmentation
# (Supplementary Material)

**Maxime Bucher**
valeo.ai
maxime.bucher@valeo.com

**Tuan-Hung Vu**
valeo.ai
tuan-hung.vu@valeo.com

**Matthieu Cord**
Sorbonne Université
valeo.ai
matthieu.cord@lip6.fr

**Patrick Pérez**
valeo.ai
patrick.perez@valeo.com

In this document, we provide qualitative examples to assess visually the behavior of ZS3Net and ZS5Net.

Figure 1: **Additional qualitative results on Pascal-VOC**. From top to bottom, results of the 2-, 4-, 6-, 8- and 10-unseen set-ups. (a) Input image, (b) semantic segmentation ground-truth, (c) segmentation without zero-shot learning, (d) results with proposed ZS3Net. Unseen classes: motorbike plane cat sofa train chair; Some seen classes: bicycle boat dog person background. Best seen in colors.

Figure 2: **Additional qualitative results on Pascal-Context**. From top to bottom, results of the 2-, 4-, 6-, 8- and 10-unseen set-ups. (a) Input image, (b) semantic segmentation ground-truth, (c) segmentation without zero-shot learning, (d) results with proposed ZS3Net. Unseen classes: cow cat boat bird plane; Some seen classes: horse dog background sky. Best viewed in colors.

Figure 3: **Additional qualitative results with self-training on Pascal-VOC**. From top to bottom, results of the 2-, 4-, 6-, 8- and 10-unseen set-ups. (a) Input image, (b) semantic segmentation ground-truth, (c) segmentation with ZS3Net, (d) result with additional self-training (ZS5Net). Unseen classes: motorbike plane cat train chair; Some seen classes: bicycle boat person background. Best seen in colors.

Figure 4: **Additional qualitative results with self-training on Pascal-Context**. From top to bottom, results of the 2-, 4-, 6-, 8- and 10-unseen set-ups. (a) Input image, (b) semantic segmentation ground-truth, (c) segmentation with ZS3Net, (d) result with additional self-training (ZS5Net). Unseen classes: cow cat sofa boat bird plane; Some seen classes: horse dog background sky. Best viewed in colors.