[Reviews · NeurIPS 2019]

Reviewer 1



In general, I like the paper: it tackles previously under-explored task and proposes a novel approach to tackle it. The paper is well-written and easy-to-follow, however I suggest improving exposition of "graph context encoding" by providing more detailed explanations or preparing illustrations. The overall approach appears to be novel to me. The main idea is to train a generative model of pixel representations that is conditioned on text embedding of seen and unseen classes. Later, authors use this model to train a classifier for unseen classes. As an additional step, authors demonstrate that pseudo-label technique can be used to further improve performance. The experiments are conducted on two challenging datasets and quantitative/qualitative results appear to be rather convincing. I am mainly concerned with the choice of baselines. Authors implement only a single baseline. In theory, it may be sufficient, considering that zero-shot segmentation learning appears to be a novel task. However, I wonder whether other classical ZSL learning techniques can be trivially adapted for ZSL segmentation and how they compare to the proposed technique.

Reviewer 2



Originality: The problem tackled is in itself original, it is well motivated (not artificial), and can open a new line of research, combining the best of semantic segmentation and zero-shot learning. The approach proposed is a straightforward adaptation of [7] to the proposed settings. Quality: Given that the problem is new, there were no real competitors, so the authors came up with a sound baseline. The main approach described in the paper is reported to work better than this baseline on Pascal VOC and Pascal-Context datasets. The choice of datasets is correct, although semantic segmentation papers tend to report results also on others such as COCO, and Cityscapes. Clarity: The paper reads well overall, and the figures (mainly Fig. 2) help in the understanding. The problem is well motivated, and the paper organization is ok. Significance: As mentioned above, other researchers may get interest in the proposed problem, trying to get better results at this. In this sense the contribution of the paper is important. The contribution about zero-shot learning and self-training (ZS5Net) seems to miss that there is a subarea within ZSL, called transductive ZSL, that has studied this same problem before, with papers such as: Zero-Shot Recognition via Structured Prediction. Zhang & Saligrama, ECCV 2016, Zero-Shot Classification with Discriminative Semantic Representation Learning, Ye & Guo, CVPR 2017. Zero-Shot Recognition using Dual Visual-Semantic Mapping Paths, Li et al, CVPR 2017 Zero-Shot Learning via Class-Conditioned Deep Generative Models, Wang et al. AAAI 2018, Transductive Unbiased Embedding for Zero-Shot Learning, Song et al. CVPR 2018.

Reviewer 3



Writing ---------- The paper is well written and well positioned wrt previous work, overall. However, the main technical core of the paper is only about 1.5 pages long. It would have been better imho to have less detailed results tables and reuse that space for explaining the technique in greater detail. Novelty ---------- While I agree that this paper is the first to address zero-shot semantic segmentation, there have been many papers on zero-shot image classification, abd also some papers on zero-shot object class detection. Importantly, the technique proposed is a direct adaptation of [7], a previous technique for zero-shot image classification. The way the authors adapt it appears to be straightforward, also because the problem of semantic segmentation is (as usual) tackled as a pixel classification problem. So, instead of one class per image, you have one class per pixel, and the authors apply [7] essentially unchanged. The other claimed novelty points in my opinion do not add substantially to the paper: (1) the self-training strategy (lines 144-152) is a classic from the semi-supervised learning literature, and is described in 9 lines in this paper. Besides, it breaks the usual zero-shot learning setting, but that's not the main point. (2) the graph-context encoding in my opinion requires an unrealistic data setup to work: you need segmentation mask annotations for the unseen classes, but without the images themselves. I cannot imagine how this could ever happen. In fact, this corresponds to full supervision for the unseen classes too. With this in mind, the authors should then compare this module of their method to previous context modeling techniques from the fully supervised semantic segmentation literature. Experiments ----------------- The results appear to be good overall, with especially big gains brought by the zero-shot self-training. There are a few methodologically subtle points though: - lines 179-188: it's great that the authors carefully tease out which of their unseen classes do not appear in the ImageNet dataset used to pre-train their models. However, these are only two classes, making their experiments with 4,6,8,10 unseen classes less valid. Moreover, I believe the right protocol would have been to simply remove all of your unseen classes from ImageNet, and then use that ablated dataset for pretraining your models. That would lead to a fully clean protocol. - as a baseline, the authors adapt a zero-shot classification approach [16] dating back to 2013. This does not seem satisfactory, given that there are many more recent zero-shot classification papers, as cited in lines 30-39. In fact, this paper itself is an adaptation of a recent zero-shot classification work [7] (2017). So there is a continuum of baselines possible, and 2013 really feels outdated. Reaction to the rebuttal ================== The rebuttal is very well written and reduces my concern on novelty a bit, as adapting [7] to pixel-level is indeed not that trivial. A second reply that helps upgrade my opinion is that the authors state that a recent study has shown that the Devise baseline [16] is anyway already very good. Moreover, they also add a more recent baseline [1]. Perhaps the core question for accepting this paper is: why is the work of adapting the zero-shot classification baseline [16] considered 'a baseline', whereas adapting another zero-shot classification work [7] is considered a contribution worthy of NeurIPS? One possible answer is that they are both too simple to grant acceptance. But another possible answer is that there is sufficient (marginal) novelty in either of them. The other replies are unconvincing: on retraining on properly ablated ImageNet, the authors essentially say 'it's too much work' and 'make training from scratch challenging'. But this is the core of the meaning of zero-shot learning! If we say 'it's too hard to evaluate in zero-shot setting in practice', then we should not work in this field. We are trying to be scientific after all. Moreover, the answer on the realism of graph context is really not convincing: the authors just say 'it's one form of prior, it could be another one'. But the point is: THIS form of prior is unrealistic. You cannot have segmented object outlines but no image pixels. In the light of the overall novelty of the task itself, the good results (at least compared to some sort of baselines), and the rebuttal, I raised my rating to 5. It would not be so bad if this paper gets in.

[Author Response · NeurIPS 2019]

We thank the reviewers for their encouraging and constructive comments. We are pleased that they find the paper well written and acknowledge the novelty and originality of the proposed task, which "has a potential to spark interest" (R1) and "may lead to future papers studying it" (R2). Regarding the proposed framework, R1 and R2 not only find it "sound" and "novel" but also stress the "re-implementation ease" from which "practitioners may benefit" (R1). Still, the reviewers raise points of improvement (R1, R3) and suggest a discussion about a related task (R2). We carefully address these comments below. Some of our answers will be included in the paper if accepted.

### Reviewer 1

**More classical ZSL baselines.** Recent works ([42] and [Schonfeld CVPR'19][1]) reported that DeViSe [15] is among the best classic techniques for generalized zero-shot learning (GZSL). Based on this, we believe that other classic methods would perform similarly to DeViSe if used instead in our GZSL semantic segmentation baseline. During the rebuttal period, we nonetheless conducted additional experiments, adapting ALE [1] for zero-shot segmentation on Pascal-VOC with $K = 2$. They yielded $68.1\%$ and $4.6\%$ mIoU for seen and unseen classes (harmonic mean of $8.6\%$), on a par with the DeViSe-based baseline in the paper. Such poor generalized-setting performance of classical ZSL methods confirm again the conclusion in [9]. Following R1's suggestion, we will include more of such classical baselines with discussion in the paper, if accepted.

**Graph context clarity.** We apologize for the lack of details on the graph context (GC). This is in part due to our initial intent to devote lots of attention and space to the experiments. In an attempt to mitigate this unbalance, we had included GC visualization in the supplementary material, which appears insufficient. If accepted, we will use the additional page at best to include more technical details and visualizations on GC.

### Reviewer 2

**Other datasets.** In state-of-the-art semantic segmentation works, Pascal VOC 2012 and Pascal Context still serve as the main benchmarks. The recent COCO-stuff dataset [Caesar CVPR'18], though larger in scale, is very similar to Pascal Context. We thus expect similar performance behaviors on it. While we have not yet completed such experiments as of now, this will be done, and would be reported in our revision of the paper. R2's suggestion of looking into urban scene datasets like Cityscape is also interesting and worth investigating in the future.

**Transductive ZSL.** While our central contribution, ZS3Net, is not transductive (no data, even unlabelled, is available at train time for unseen classes), the ZS5Net variant indeed appears related to transductive zero shot learning. We thank the reviewer for bringing this to our attention. Apart from the fact that referred papers concern all image classification, another difference is worth mentioning though: all but [Song CVPR'18] consider purely transductive settings where *all* unseen class samples are already available at training time. By contrast, our ZS5Net learns from a mix of labelled and unlabelled training data, and is evaluated on a different test set (effectively, a form of semi-supervised learning).

### Reviewer 3

**Novelty.** Being the first to address zero-shot semantic segmentation, we naturally built on existing zero-shot learning literature. Yet, as abundantly exemplified in fully supervised learning, moving from image-level categorization to pixel-level recognition is not as direct or straightforward as it might seem. Highly structured prediction remains a challenge, which we revisit in the context of zero shot learning. While previous generative-based ZSL methods like [7] operate on image-level features, our generator operates on pixel-level ones. Moreover, to encode spatial context, we propose a novel graph convolutional generator which, conditioned on context graphs, generates corresponding structured pixel-level features. Also, as we shall clarify, our framework is not solely bound to GMMN as in [7]; it is in fact agnostic to the choice of the generative model. For instance, we experimented a variant of ZS3Net based on GAN [42], which turned out to be on a par with the reported GMMN-based one. In the submission, GMMN was chosen due to its better stability. In the end, ZS3Net achieves promising, quantitative and qualitative results on a never addressed task, and its ZS5Net extension yields performance very close to the full-supervision upper-bound.

**Retraining on ImageNet.** We acknowledge R3's suggestion of re-training ResNet-101 only on seen classes images. Actually, this should be the *de facto* protocol for all zero-shot learning works, to avoid supervision leakage from unseen classes. Our main concern is the challenge of such an undertaking: beside mere time and compute requirements, the absence of current reference performance with such a setting might make training from scratch even more challenging; this might also raise fair comparison issues with future works in the field. Anyhow, we will try our best to overcome these challenges and to extend our manuscript accordingly.

**Realism of graph context.** One who has never seen a 'zebra' can still learn from the fact that zebras live in African treeless grasslands. Such a coarse context prior is actually enough to construct a valid context graph in our approach. Indeed, the way we design this graph is in fact very loose, requiring only relative spatial arrangements, not object shapes (as illustrated in Fig. 2 of supplementary). Using segmentation masks is only one possible strategy, which we chose for the sake of convenience. However, any other, less precise contextual descriptions of unseen objects would suffice to build useful graphs. Anyhow, we would argue that even using segmentation masks for that purpose does not amount to full supervision for the unseen classes since images themselves are not accessible.

**Baselines.** We kindly refer R3 to our first answer for R1 above.

## Footnotes

[1]Schonfeld *et al.*, Generalized zero-and few-shot learning via aligned variational autoencoders, CVPR 2019


[Meta-Review · NeurIPS 2019]

The submission originally received mixed scores. The reviewers acknowledged that the submission is the first work that addresses zero-shot semantic image segmentation, but they criticized the novelty compared to existing classification methods and identified shortcomings of the experimental setting. The reviewers appreciated the author response and subsequently discussed the work in detail. However, the conclusion was that despite some shortcomings, the community will potentially benefit from the work and that the submission should be accepted. The authors are strongly encouraged to make their results understandable and reproducible, e.g. by releasing code, complete outputs and ideally pretrained models.